# Data Acquisition for Estimating Energy-Efficient Solar-Powered Sensor Node Performance for Usage in Industrial IoT



Dalibor Dobrilovic [1], Jasmina Pekez [1,*], Eleonora Desnica [1], Ljiljana Radovanovic [1], Ivan Palinkas [2], Milica Mazalica [1], Luka Djordjević [1] and Sinisa Mihajlovic [1]

[1] Technical Faculty "Mihajlo Pupin" Zrenjanin, University of Novi Sad, 23000 Zrenjanin, Serbia; dalibor.dobrilovic@tfzr.rs (D.D.); eleonora.desnica@tfzr.rs (E.D.); ljiljana.radovanovic@tfzr.rs (L.R.)

[2] Technical College of Applied Sciences, 23000 Zrenjanin, Serbia; ivanpalinkas@gmail.com

* Correspondence: jasmina.pekez@tfzr.rs

**Abstract:** In the era of rapid technological growth, we are facing increased energy consumption. The question of using renewable energy sources is also essential for the sustainability of wireless sensor networks and the Industrial Internet of Things, especially in scenarios where there is a need to deploy an extensive number of sensor nodes and smart devices in industrial environments. Because of that, this paper targets the problem of monitoring the operations of solar-powered wireless sensor nodes applicable for a variety of Industrial IoT environments, considering their required locations in outdoor scenarios and the efficient solar power harvesting effects. This paper proposes a distributed wireless sensor network system architecture based on open-source hardware and open-source software technologies to achieve that. The proposed architecture is designed for acquiring solar radiation data and other ambient parameters (solar panel and ambient temperature, light intensity, etc.). These data are collected primarily to define estimation techniques using nonlinear regression for predicting solar panel voltage outputs that can be used to achieve energy-efficient operations of solar-powered sensor nodes in outdoor Industrial IoT systems. Additionally, data can be used to analyze and monitor the influence of multiple ambient data on the efficiency of solar panels and, thus, powering sensor nodes. The architecture proposal considers the variety of required data and the transmission and storage of harvested data for further processing. The proposed architecture is implemented in the small-scale variants for evaluation and testing. The platform is further evaluated with the prototype sensor node for collecting solar panel voltage generation data with open-source hardware and low-cost components for designing such data acquisition nodes. The sensor node is evaluated in different scenarios with solar and artificial light conditions for the feasibility of the proposed architecture and justification of its usage. As a result of this research, the platform and the method for implementing estimation techniques for sensor nodes in various sensor and IoT networks, which helps to achieve edge intelligence, is established.

**Keywords:** solar radiation data; solar data harvesting; wireless sensor networks; solar-powered sensor nodes; data acquisition; industrial IoT; solar panel efficiency

## 1. Introduction

The rapid development of technology and equally rapid growth of the world population caused the problem of energy sources and their exploration [1,2]. The issue is even raised with the development and increasing deployment of the outdoor Industrial Internet of Things (IIoT) and smart technology systems [3,4]. As a result, the question of using solar energy as one of the most considered and applicable renewable energy sources is essential, especially in the IIoT and smart technology systems [5]. It is caused by the broad implementation of sensor networks but also by the need for a large number of sensor nodes and smart devices used in these systems.

These factors influenced the importance of using solar-powered sensors and smart devices and caused a variety of research related to this issue. One of the crucial aspects of using solar energy for powering sensor nodes is the extension of the operation time of these devices. So, to optimally utilize solar-powered sensor nodes in given situations, it is vital to identify the most suitable locations for node deployment and, therefore, to estimate solar panel outputs in specific locations. This helps to model sensor node operations and create adaptive energy-saving modes depending on the estimated outputs. A variety of approaches are used in the described process. This paper proposes an approach based on a wireless sensor network for collecting data on solar radiation and using different types of sensors to estimate solar panel performance, especially in outdoor IIoT scenarios.

This research considers the problem of estimating the energy supply of solar-powered wireless sensor networks for various IIoT environments. The system proposed in this paper is developed to collect data that can be used with nonlinear regression techniques for solar panel output estimation. The estimated outputs can be further used for estimating optimal energy-efficient operation modes of solar-powered sensor nodes, thus achieving efficient solar power harvesting effects. As a solution, this paper presents the distributed wireless sensor network system architecture based on open hardware and open-source technologies designed to acquire solar radiation data and other ambient parameters. The acquisition of solar radiation data and other ambient parameters (panel and ambient temperature, light intensity, etc.) strongly depends on the design and platform efficiency of sensor nodes deployed in this network.

The nodes are designed to collect solar radiation-related data, such as the current and voltage generated by the solar panel and ambient data. These ambient data are collected for further processing and analysis of the influence of ambient on the efficiency of solar panels and solar-powered sensor nodes. The architecture also considers transmission and data storage of collected data for further processing. The presented system is implemented in a small-scale variant for testing and evaluation. It is implemented as a prototype of a reduced configuration sensor node with sensors for measuring the voltage, visible and ultraviolet (UV) light, solar panel temperature, air temperature, and humidity. To avoid long-term testing, we tested the prototype in various scenarios with natural solar and artificial light to prove the feasibility of such nodes in the proposed architecture and justify using open-source hardware and low-cost components to design such data acquisition nodes.

The contribution of this paper is the proposed model for solar radiation data acquisition and the approach to using these data to estimate solar panel efficiency that can be used to model the operations of solar nodes to achieve energy efficiency and extend the node lifecycle. The contribution can be further used to implement estimation techniques for sensor nodes in various sensor and IoT networks, which helps achieve edge intelligence not limited to solar data acquisition networks. The model is motivated by the goal of combining the two different approaches in solar radiation estimation. These two approaches are systems designed to monitor solar panels and other electrical systems with the use of sensor nodes as a single device or as a part of a distributed sensor network with the projects designed to estimate solar radiation using meteorological, GIS (Geographic Information System), LIDAR (Light Detection and Ranging) data, terrain configuration, and satellite imagery. These two approaches motivated the design of the sensor network for acquiring solar radiation data for solar panel output estimation.

The model is presented with a fully equipped solar-powered sensor node, which can be used in various sensor networks and with the architecture of the network. The model of a reduced-equipped sensor node is also presented. The reduced model of the sensor node is evaluated for usage in a sensor network where the collected data will be used for solar panel outputs.

The model is partially evaluated with the efficiency and feasibility of using the low-cost open-source hardware components for developing sensor nodes for solar panel data acquisition. The reduced-equipped sensor node is used for this evaluation. The experiment is not made in the long-term period. Instead, the experiment is performed under the

simulated environment with direct solar light, reduced solar light, indoor ambient light, and different artificial light sources at different distances from the solar panel. In addition to standard light in the experimental room, incandescent, compact fluorescent lamps (CFL), and light-emitting diode (LED) bulbs are used as additional light sources. The accuracy of a low-cost voltage sensor is compared with a digital multimeter. Next, the relation of measured values with other sensors, such as UV and visible light, and the panel temperature is analyzed. The measurement results of using open-source hardware in sensor networks to acquire solar panel data and to model the energy-efficient operations of deployed solar nodes.

Considering the importance of solar power generation, the potential expansion of solar-powered sensor networks in the future, and the significance of better managing solar-powered sensor nodes, this system can be beneficial. The system can effectively forecast the output of sensor-based solar panels. Thus, combined with the data analysis techniques, it can be used to determine the most energy-efficient mode depending on the panel output and energy supply. Additionally, the research results introduce the methodology for implementing estimation techniques on sensor nodes in various data acquisition networks.

This paper is structured as follows. After the introduction, the related work with the motivation, similar research, and recent achievements in the field is discussed. Section 3 presents the architecture of the sensor network for solar panel data acquisition. In Section 4, the experiment for validating the sensor node prototype as a vital element of the presented architecture is conducted, followed by the result analyses in Section 5. The concluding remarks and possible further research directions are discussed at the end of the paper.

## 2. Related Work

There is a variety of related research that motivated the research presented in this paper. In summary, the first group of related research are systems designed to monitor solar panel plants and other electrical systems, current and voltage parameters, and use sensor nodes as a single device or as a part of a distributed wireless sensor network. The second group of related works uses meteorological data, satellite imagery, terrain configuration, and LIDAR data to make maps and models for estimating solar radiation and potential locations for primarily building rooftop solar panels. Our motivation was to integrate those two approaches.

The related research group in the field of using sensor nodes for solar panel performance and energy consumption monitoring is presented in Table 1 with all relevant characteristics. The column named difference points out the difference between previous and present works. The research presented in this paper targeted the gaps that needed to be covered in enlisted works.

**Table 1.** Comparative view of solar and energy monitoring systems.

| Ref. No. | Dist. | Connectivity | Purpose | Platform | Key Outcomes | Difference |
|---|---|---|---|---|---|---|
| [6] | No | UART bus | A low-cost solution for real-time instrumentation of the photovoltaic (PV) panel characteristics such as voltage and current power. | Arduino UNO PLX-DAQ data acquisition Excel Macro | confirmation of the effectiveness of the developed virtual instrumentation system | single point of acquisition, no networking and distribution |
| [7] | No | Bluetooth HC-05 | Smart voltage and current monitoring system (SVCMS) for monitoring a three-phase electrical system with three voltage and current sensors. | Arduino Nano V3.0 Android Smartphone | Android smartphone application for monitoring the voltage and current measurements | single point of acquisition, no networking and distribution |

**Table 1.** *Cont.*

| Ref. No. | Dist. | Connectivity | Purpose | Platform | Key Outcomes | Difference |
|---|---|---|---|---|---|---|
| [8] | Yes | Zigbee USB | A prototype of a low-cost home energy management system (HEMS). This platform aims to monitor the energy consumption of typical household devices so that the users can access the consumption of each device separately and, in the end, establish a strategy that allows them to reduce energy consumption at home. | Arduino Uno | low-cost home energy management system | single-type sensor, short-range indoor wireless connection |
| [9] | No | Wired USB | Data Acquisition System (DAS) is designed to collect voltage and current data in real-time at variable load resistance during an experimental characterization analysis of a $3 \times 3$ size photovoltaic (PV) system under partial shading (PSC) conditions using Analog voltage and current sensors. | Arduino Nano | collecting voltage and current data in real-time | single point of acquisition, wired connection to PC |
| [10] | No | HC-05 Bluetooth module | The system for monitoring the robotic base and its output voltages. The measuring system uses an Arduino microcontroller, current ACS712, and voltage sensor FZ0430. | Arduino UNO Arduino SmartPhone | low-cost monitoring of the robotic base and its output voltages | Single point of acquisition, short-range wireless connection |
| [11] | No | Wi-Fi IEEE 802.11ac | The energy consumption characteristics monitoring for robots with INA219 high-side current sense amplifier to capture power, current, and voltage measurements. | Raspberry Pi4 model B | energy consumption monitoring for robots | single point of acquisition, mobile wireless connection |
| [12] | Yes | LoRa/ LoRaWAN | Monitoring PV system-related parameters (voltage, current, power, energy, light intensity, temperature, and humidity) and updating this information to the cloud. Data are sent to the LoRaWAN gateway and further to The Things Network (TTN). | Arduino UNO | cloud PV system-related parameters monitoring | PC-centric acquisition system, no PV performance dependency analyses, different architecture |
| [13] | Yes | LoRa/Wi-Fi | Monitoring climatic variables and photovoltaic generation for Smart Grid application (voltage, current, alternating power, and seven meteorological variables). | HeltecWi-Fi LoRa 32 (V2) IoT dev-board | climatic and PV monitoring for the Smart Grid application | PC-centric acquisition system, no PV performance dependency analyses, different architecture |
| [14] | Yes | I2C, SPI, Serial, proprietary NRF24L01 | Supervisory Control and Data Acquisition system for a microgrid testbed. | Arduino UNO/ Raspberry Pi | data acquisition system for a microgrid | proprietary communication, the limited number of wireless nodes |
| [15] | No | USB, Wi-Fi, 3G/LTE/4G, etc.), Ethernet | Open source, low-cost, precise, and reliable power and electric energy meter and power quality analyzer for homes in urban or rural areas. | open meter custom build board | low-cost, electric energy meter and analyzer | no network architecture defined |

**Table 1.** *Cont.*

| Ref. No. | Dist. | Connectivity | Purpose | Platform | Key Outcomes | Difference |
|---|---|---|---|---|---|---|
| [16] | No | Ethernet | Monitoring system based on open-source hardware and software for tracking the temperature of the photovoltaic generator in an SMG. | Arduino MEGA 2560 R3 RPi model 3 ver. B | the temperature of the photovoltaic generator monitoring | single point of acquisition, wired connectivity |
| [17] | No | Modbus-RTU, TCP/IP, and Wi-Fi | PV Monitoring System for a Water Pumping Scheme to provide a valuable tool for the operation, management, and development of these facilities. | Raspberry Pi ADAM 4017+ | PV monitoring | single point of acquisition, wired connectivity |
| [18] | Yes | 433 MHz RF HopeRF RFM69CW | Low-cost PV-module monitoring system based on open-source solutions for monitoring installations at the PV-module level, giving detailed information regarding PV power-plant performance (monitoring PV module and meteorological data) | Arduino UNO Raspberry Pi | PV monitoring | no network architecture defined |
| [19] | Yes | Wi-Fi | System for real-time cloud monitoring of a decentralized photovoltaic (PV) plant with ACS712 current sensor, LM35 temperature sensor, LP02 pyranometer, and DHT11 sensor. | Raspberry Pi ADCES (SanUSB board) | cloud PV monitoring | no PV performance dependency analyses, different architecture |
| [20] | Yes | Wi-Fi | IoT modular system to compose a worldwide monitoring network focused on meteorological and PV modules temperature data (PV modules temperature, meteorological data such as solar irradiance, ambient temperature, relative humidity, and wind speed) | ESP8266 ESP32 | photovoltaic plants monitoring | no PV performance dependency analyses, different architecture |
| [21] | No | Modbus TCP/IP and OPC communication protocols | real-time supervision and predictive fault diagnosis of solar panel strings | ESP8266 module, ASC712-5A, and FZ0430 sensors and relay modules | predictive diagnosis method, based on online detection centered on each solar panel of the PV string | wired centralized standalone system |
| [22] | No | Bluetooth | Photovoltaic tracking system and cleaning connected to a smart board | Arduino Mega2560, current sensor and voltage sensor | IoT-based smart household distribution board to monitor the functioning of various appliances | Short-range communication, standalone system |
| [23] | No | Wi-Fi | Decentralized, low-cost alternative Automatic Data Acquisition Systems (ADAS) based on the ESP32 microcontroller | ESP32, sensors (Kipp and Zonen CMP11, PT100, Biotech VZS-007, SHT20 waterproof) | Data acquisition system for solar thermal collector testing | Standalone monitoring system |

The second group of research related to this one is focused on mapping, analyzing, and estimating rooftop solar potential in urban environments. This research is conducted to estimate the feasibility of rooftop solar panels for residential building energy consumption. Table 2 summarizes representative research analyzed in the preparation phase of the paper.

The table gives an overview and brief description of the methodology and source data used for estimation.

**Table 2.** Comparative view of solar estimation and mapping systems.

| Ref. No. | Description | Data Used | Artificial Intelligence |
|---|---|---|---|
| [24] | methodology for estimating the solar potential with the generation of a 3D structure | height data and roof types using satellite imagery and simulation of shadows | No |
| [25] | determination of energy potential of rooftop solar PVs | building height model | No |
| [26] | method for selecting suitable locations for installing solar panels | Graphic Processing Unit (GPU)- solar radiation model SHORTWAVE-C for simulation of direct and non-direct solar radiation | No |
| [27] | a solar irradiation estimation solution for three-dimensional (3D) cities | annual irradiations on urban envelopes | No |
| [28] | identification of the building roofs for estimating the city's solar potential | U-Net of deep learning technology in combination with satellite maps | Deep learning |
| [29] | 3D solar potential model | Light Detection and Ranging (LIDAR) data rendered in the ArcGIS platform using CityEngine | No |
| [30] | hybrid models for solar radiation using | meteorological data | multiple linear regression, neural network, and random forest |
| [31] | artificial intelligence-based solar radiation estimation model for green energy utilization | | Artificial Neural Networks (ANN), Support vector machine (SVM), Random Forest (RF) |
| [32] | prediction of the performance of solar collectors | experimental data collection | clustering analysis with the Back Propagation (BP) and Convolutional Neural Network (CNN) models. |
| [33] | solar radiation prediction | meteorological data | ANN model, and a recurrent neural network (RNN) model |
| [34] | mapping clear-sky surface solar ultraviolet radiation | Chinese Ecosystem Research Network (CERN) stations measurements | Machine learning |
| [35] | solar radiation estimation | CAMS Radiation Service solar radiation data | Ordinary Kriging and distance weighting, non-supervised competitive ANN-Self Organizing Map |

Our solution combines the experiences and goals of both groups and differs significantly. It uses the first group of referred works as motivation and examples for designing the data-collecting platform and the distributed sensor node network. The second group of referred works is used only as a role model for estimating solar radiation potential. Unlike the presented works in Table 2, our proposal used nonlinear regression to discover the sensor that can be used to predict the solar power output with the best possible accuracy.

In addition to designing a sensor network, our work covers the design of sensor nodes for a distributed wireless sensor network for solar panel data acquisition. The sensor

network is built on nodes with a minimum set of components. The system is designed to collect data that will be used further for solar panel output estimation and modeling energy-efficient sensor nodes.

## 3. Architecture of Sensor Network for Solar Radiation Data Acquisition

The architecture of the fully equipped sensor node for solar panel data acquisition is presented in Figure 1. The system is designed to collect solar panel and ambient data. The solar panel data consists of solar panel current and voltage data logging. The current and the voltage generated from the solar panel and the current and voltage used for charging the battery are essential for monitoring. The ambient data, such as solar panel surface temperature, air temperature and humidity, light intensity, dust, and rain detection, are also interesting for monitoring. The purpose of monitoring ambient data is to analyze the impact of these side factors on solar panel energy generation.

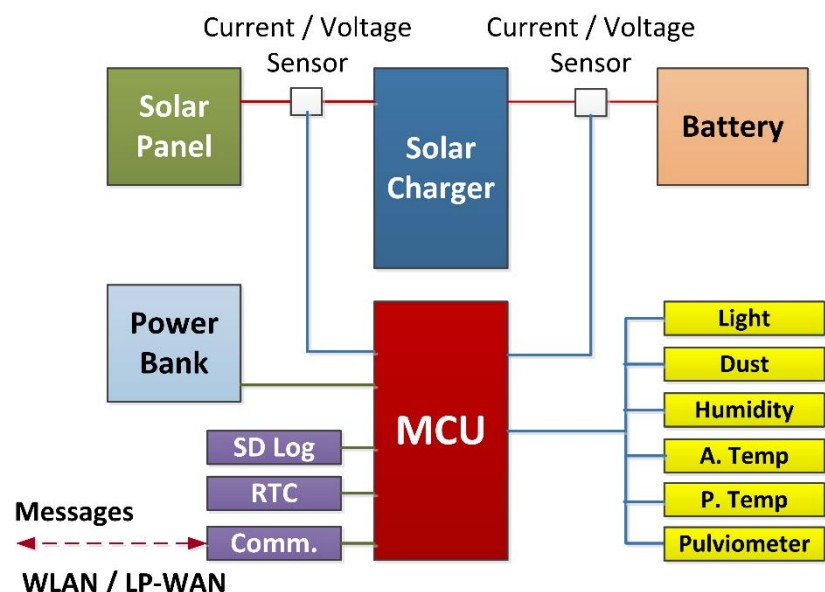

**Figure 1.** The architecture of a fully equipped wireless sensor node for solar data acquisition.

In the model, all sensors are connected to the same microcontroller unit (MCU), as shown in Figure 1. The MCU can be Arduino UNO or Arduino MEGA. In addition to sensors, the MCU can be connected to Real Time Clock (RTC), communication module, and SD card module for data logging and a liquid–crystal display (LCD) or organic light-emitting diode (OLED) module optionally. The communication module depends on the coverage of the presented network, which will be discussed in more detail in the following part of this section.

The possible set of sensors that can be considered for the proposed sensor node is the DHT-11 or DHT-22 sensor for air temperature and humidity, TMP36 for a solar panel temperature, BH1750 sensor and other light sensors for light intensity measuring; UV sensor; ACS712 or INA169 current sensor; MAX471 current and voltage sensor; etc. The solar charger is Seeedstudio Li-Po Rider Pro. It is used for managing Li-Po battery 500 mAh charging and discharging. The solar panel dimensions range from 160 mm × 138 mm × 2.5 mm to 130 mm × 87 mm × 2.5 mm, but other sizes will be taken into consideration also. The panel's efficiency is 16%, voltage 5.5V, 3W power, and peak current depends and ranges from 540 mA to 270 mA. The communication module for Wi-Fi technology can be ESP8266 or ESP32. It is interesting to consider the usage of Wemos D1 R2 or NodeMCU boards with integrated ESP8266/ESP32 modules instead of using Arduino UNO boards. The limiting factor in using Wemos or NodeMCU boards can be the possibility of connecting only one analog sensor.

The proposed fully equipped sensor node should be considered part of the distributed wireless sensor network for solar panel data acquisition, as presented in Figure 2. The sensor network has nine elements. It is based on the smart factory sensor network prototype shown in [36].

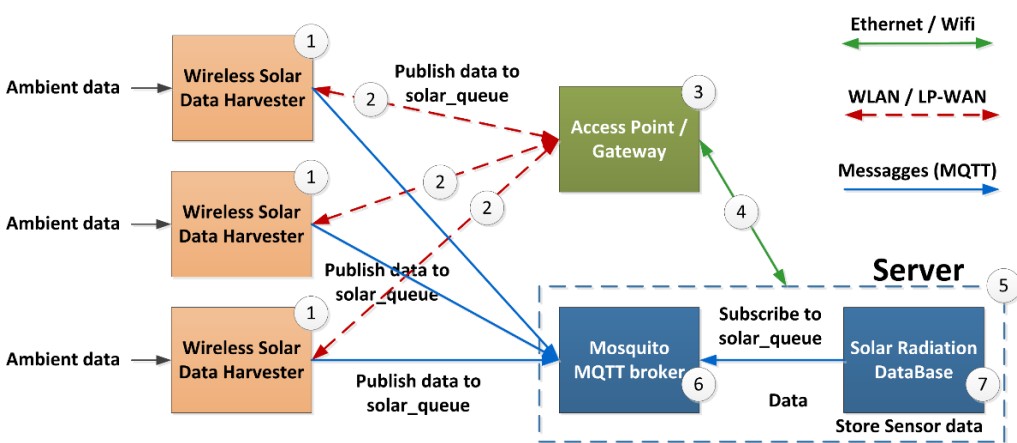

**Figure 2.** Elements of distributed wireless sensor network for solar radiation data acquisition.

Wireless sensor nodes are the first element of the distributed sensor network (1). The wireless sensor network coverage defines the technology used for communication (2). The Wi-Fi technology and one of its variants (IEEE 802.11b/h/n/ac/ax/ad) should be considered if the network is designed for short or middle-range coverage. For long-range coverage (a few kilometers), the LoRa/LoRaWAN should be considered [37–39]. For further research, the investigation of using the IEEE 802.11ah should be very interesting.

The same is with the device connecting the wireless nodes (3). When using one of the Wi-Fi technology variants, the access point should be used as a concentration device. When using LoRa/LoRaWAN technology, the LoRaWAN gateway should be used. The connection (4) between the server (5) and concentrating device (3) can be realized with Wi-Fi or wired technology but should be based on IP protocol. The server's primary role (5) is collecting, storing, processing, and analyzing collected data. Collected and analyzed data will be available to the system users through web interfaces with live graphs, dashboards, and reports. The collector for the data sent from the wireless nodes is the open-source server (6) based on messaging protocol, probably MQTT (MQ Telemetry Transport) [40]. Finally, data collection should be realized with the DB storage system (7), SQL, or NoSQL, depending on the number of nodes and required data acquisition and writing frequency.

The last two elements of the system depend on the communication technology used for connecting nodes (1) and concentrating device (3) and concentrating device and server (5). Suppose the connection from the node is based on Wi-Fi technology, and the further connection to the server is based on Wi-Fi or Ethernet (wired) technologies. In that case, the messages containing collected data will be sent with the MQTT messaging protocol. When LoRa/LoRaWAN protocol is used for node connectivity, the conversion from LoRa to MQTT messages will be implemented at the LoRaWAN gateway–concentrating device (3).

In this research, the authors did not test the proposed system on a full scale during the continuous operation of Arduino and clone devices. As the following sections explain, the node prototype is built as a small and laboratory-scale setup for testing purposes, as planned in the project's current phase. The future phase will cover building the sensor node network and its full data acquisition operation.

Estimating the efficiency of the proposed device, the authors rely on previous personal experiences such as in [36–39] and numerous external research sources where the Arduino UNO and its clones are proven reliable devices. The effects of high-temperature exposure of sensor nodes will be examined in the project's future phases, relying on previous

experiences. Possible measures for reducing heat and dust exposure will be used in designing protective cases for sensor nodes in further research.

The lack of experimentation with the continuous operation of Arduino boards in the open environment under direct exposure to sunlight and temperature is the major problem of this research in the current phase. However, this will be explored in many details in the future, primarily with the design of protective cases and node temperature monitoring with thermal cameras or temperature sensors.

## 4. Experiment

This section presents an experimental platform to examine the efficiency of using open-source, low-cost, and prototyping components to develop a solar panel data acquisition network. The platform's photo is shown in Figure 3. The platform is built upon the prototyped node from the previous chapter (Figure 1).

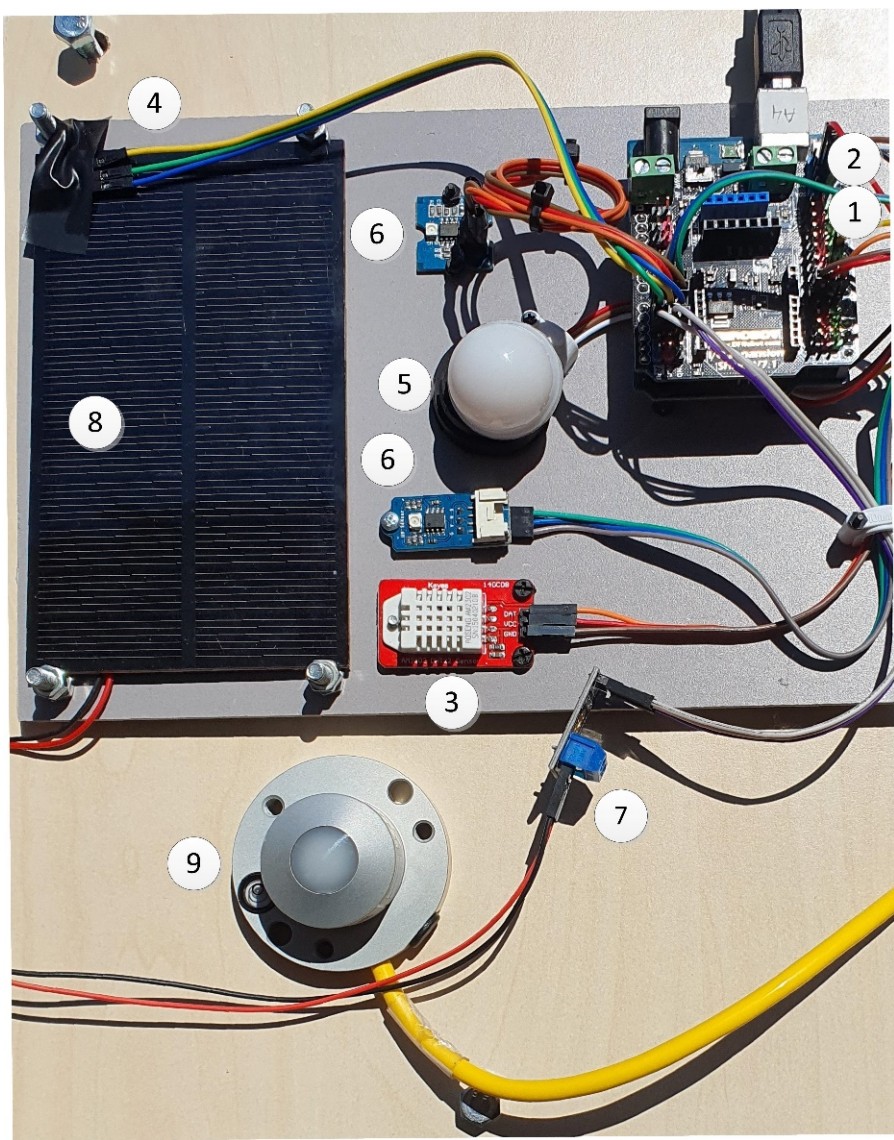

**Figure 3.** Prototype platform based on Arduino for experimenting with solar panel data acquisition.

The architecture of the testing platform is developed as a reduced model presented in Figure 1, shown in Figure 3. The platform is designed to measure the accuracy of a low-cost voltage sensor for solar panel performance monitoring. After justifying its accuracy, the relation of other measurement values with the panel voltage output is analyzed. The

voltage sensor was chosen for this experiment as the single sensor that can simultaneously measure and analyze solar radiation and solar panel efficiency.

The testing platform (Figure 3) is based on low-cost open-source hardware (OSHW) components. Open-source hardware allows rapid assembly, development, and configuration of sensor devices. The value of a sensor device built with open-source components is given in Table 3, which is below 50 EUR. Instead of Node MCU, Wemos D1 or Arduino and its clones with integrated or connected Wi-Fi modules can also be considered. If some components are changed, the price will be close to the presented one. The level of prices can depend on a region or the current offer of the market, but it can be similar.

**Table 3.** The price of the components of the sensor node.

| No. | Item | Price (EUR) |
| --- | --- | --- |
| 1 | Voltage sensor | ~2.50 |
| 2 | NodeMCU v3 | ~5.00 |
| 3 | Solar panel 137 × 81 cm | ~8.00 |
| 4 | Solar charger | ~11.00 |
| 5 | Li-Po 3.7 battery 4000 mAh | ~11.00 |
| 6 | UV sensor | ~8.00 |
| 7 | BH 1750 | ~2.00 |
| 8 | Case | ~2.00 |
| | Total | ~49.50 |

The central part of the platform is a PC (Personal Computer) with the Windows operating system. PC logs the data collected with a sensor node and digital multimeter. A Digital multimeter is used to make control measurements and validate the voltage sensor's accuracy. The MCU used in the testing platform is Arduino UNO (1). Arduino UNO is connected to the sensor shield (2). Arduino sends data to the PC via Universal Serial Bus (USB). It is programmed to read the data from the voltage sensor and send data to the PC via USB. PC reads the data sent to the COM port and logs it in comma-separated values (CSV) file using putty terminal software.

The control measurement is made with a digital multimeter (DMM) connected to a PC via USB. The data are sent to a PC, where the open-source digital multimeter reader program UltraDMM is used to visualize and log the data. Both devices are connected in parallel with the solar panel. The other components of the platform are (Figure 3): (3) DHT-22 sensor for air temperature and humidity, (4) TMP36 sensor for solar panel temperature, (5) light BH1750 sensor for light intensity, (6) UV sensors, (7) voltage sensor, (8) solar panel: 130 mm × 87 mm × 2.5 mm (efficiency 16%, 5.5V, 3W, peak current 270 mA), and (9) pyranometer. The pyranometer is not integral to this open-source platform and is only used for control measurements.

The testing phase should be before the project's continuation and the sensor network's development and deployment to prove its ability and efficiency and justify its usage. To measure the solar panel performance under different conditions in the shortest period possible, the experiment uses natural sunlight in combination with artificial light sources in indoor space. Three light bulbs are used: incandescent, compact fluorescent light (CFL), and LED. Additionally, the indoor ambient light is used (Figure 4). As described before, the response of the solar panel, the voltage change, is measured in parallel with the Arduino platform voltage sensor and digital multimeter.

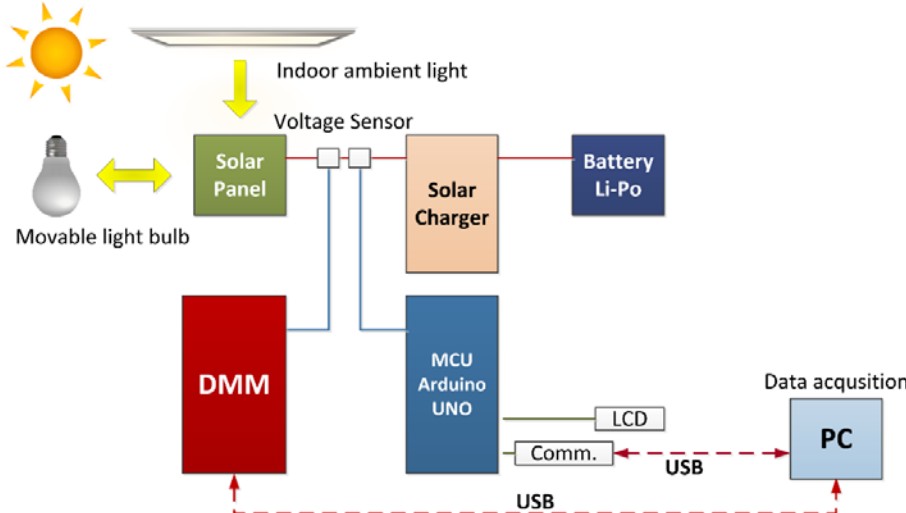

**Figure 4.** Prototype platform based on Arduino for testing the performance of voltage sensors.

The methodology of usage of the platform in the research is depicted in Figure 5, where the algorithm of the research methodology is presented. The algorithm shows seven steps. The steps are as follows: (1) design of the platform with the definition of the components and building of the platform, (2) data acquisition of the solar radiation-related data and logging to the PC, (3) parsing of collected values and (4) data processing performed at PC with the usage of Python and/or GNU Octave, (5) creation of reports and comparison of data processing outputs, (6) evaluation of the accuracy of estimation and deciding if the evaluation results are acceptable, and in the case, if they are not acceptable triggering re-design of the platform, and in the case if results are acceptable, forwarding to (7) implementation of the estimation techniques to the firmware of sensor nodes. The algorithm of the research flow is presented in Figure 5.

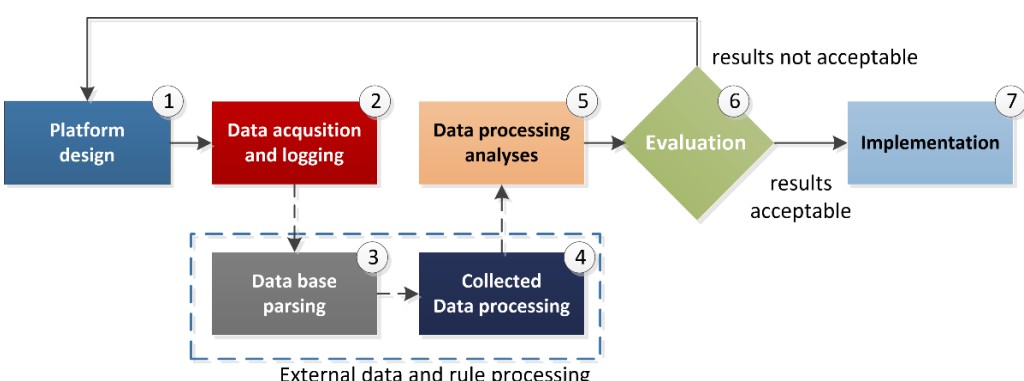

**Figure 5.** Algorithm of research flow.

## 5. Results

In this section are described the results of the experiments. The results are acquired with the experiment conducted in two phases. The experiment's first phase evaluates the voltage sensor's accuracy. The first phase is divided into two subphases. The second experiment is driven to find the relation between voltage and other measured values, such as panel temperature, air temperature and humidity, UV, and sunlight intensity.

### 5.1. Results of the Voltage Sensor Measurements

The first subphase of the first experiment is conducted with sunlight only. First, the testing platform is exposed to direct sunlight. The transparent plexiglass is placed over the testing platform in the next subphase. The amount of sunlight is reduced by gradually

covering the plexiglass with colored semi-transparent foils, as shown in Figure 6. This step-by-step covering is performed because the authors wanted to test the voltage sensor response with different sunlight intensities, avoiding testing in mid or long-term periods. By adding semi-transparent foils, the authors simulated the reduction of sunlight and different sunlight intensities. At the final stage of the testing process, the solar panel is covered with a non-transparent cover.

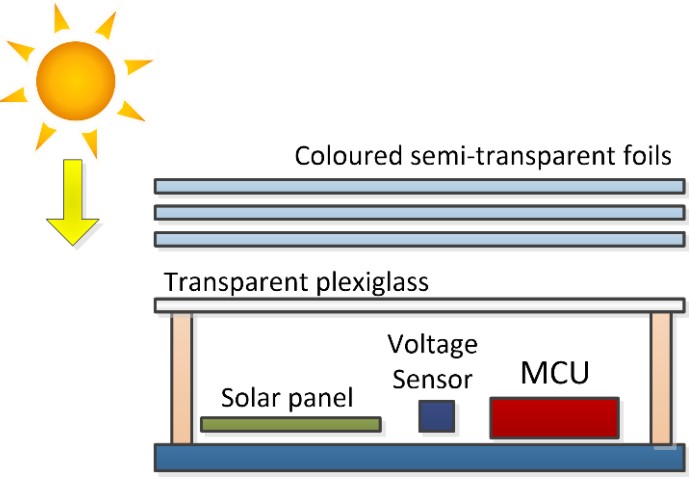

**Figure 6.** Solar panel voltage measurement with semi-transparent folios.

The effects achieved in this phase can be seen in Figure 7, with the graduate decreasing solar panel voltage. The figure shows the comparison of sensor and digital multimeter results. The voltage sensor offers good capabilities because both measurements were similar, with a slight difference. The lowest values in Figure 7 are for the test cases when a non-transparent cover is used. Even in this case, there is a distance between the solar panel and the cover, and the solar panel generated voltage.

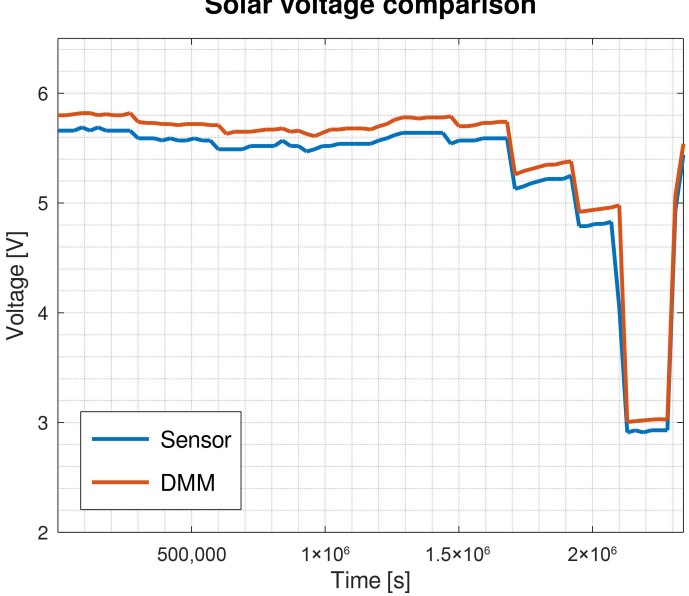

**Figure 7.** Comparison of the voltage sensor and digital multimeter measurement using sunlight.

Next, the test continued with indoor ambient light and artificial light sources. This testing phase aimed to monitor the voltage sensor response when exposed to different light sources. The testing platform is exposed to indoor ambient light and an additional

three light bulb sources to achieve that goal. Three bulbs are used: incandescent, compact fluorescent light (CFL), and light-emitting diode (LED). The bulbs are mounted to the bulb holder with an adjustable bulb height. The bulb height gradually changed as it was measured at six different distances (35, 30, 25, 20, 15, and 10 cm) from the solar panel. The seventh test phase is with the bulb turned off. The platform with the adjustable bulb holder is presented in Figure 8.

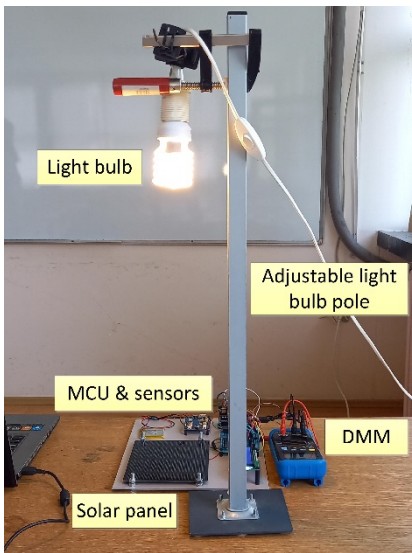

**Figure 8.** The platform with the adjustable bulb holder.

The joint results of the sensor and digital multimeter for all three bulbs are presented in Figure 9. Again, the results of both sensors are similar.

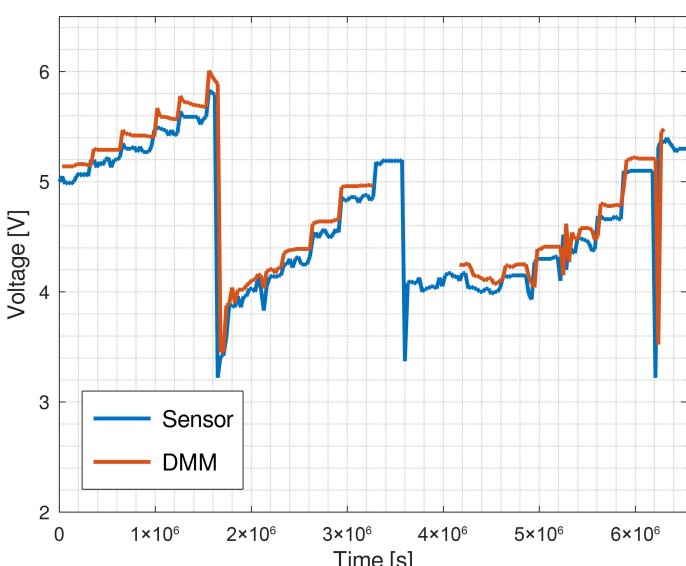

**Figure 9.** Comparison of the voltage sensor and digital multimeter cumulative measurements for three types of bulbs.

Figure 9 shows that three light sources have different values gradually. The first part of the graph is with the results of incandescent light. When the light bulb is positioned at 35 cm, the values are around 5 V, with 30 cm height values around 5.2 V, and up to 6 V

when the bulb is 10 cm from the solar panel. When the bulb is off, the measured values are around 3.2 V.

The next part of the graph shows results with the exact distances for CFL and LED bulbs, respectively.

Further, the comparative results of measurements for three bulbs separately are shown. Figures 10–12 show the comparative results for incandescent, CFL, and LED bulbs, respectively.

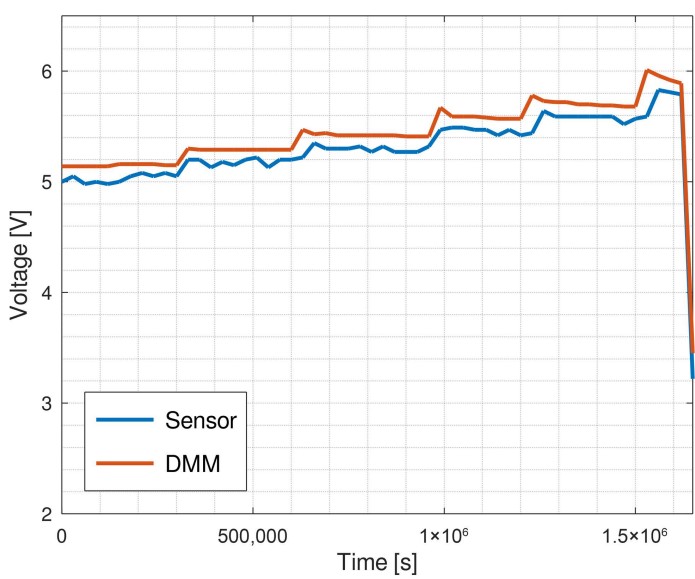

**Figure 10.** Comparison of the voltage sensor and digital multimeter measurement using an incandescent bulb.

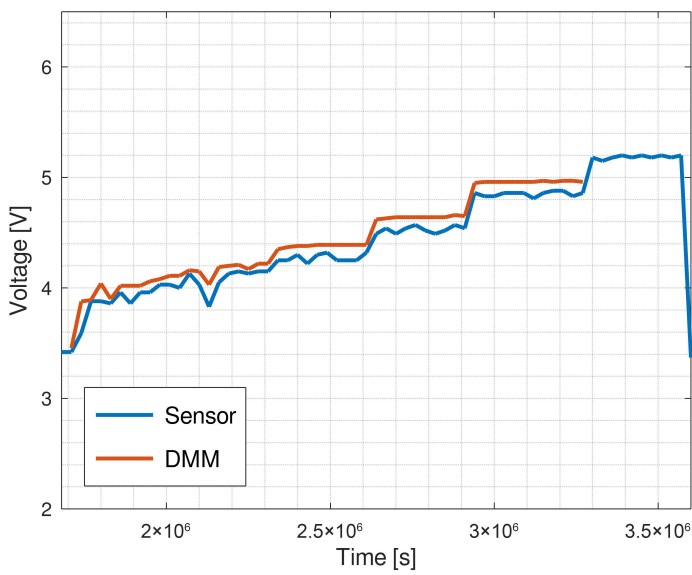

**Figure 11.** Comparison of the voltage sensor and digital multimeter measurement using CFL bulb.

**Figure 12.** Comparison of the voltage sensor and digital multimeter measurement using LED bulb.

Figure 10 compares DMM and voltage sensor measurements with the incandescent bulb used to simulate the light source. Similar measurement values are visualized. The visualization of the measurements clearly shows the voltage sensor's accuracy, confirming its justification for usage in the measurements platform.

Figures 11 and 12 similarly compare DMM and voltage sensor measurements with the CFL and LED bulbs that simulate the light source. As in the previous case, the similarity of the measured results is obvious. The visualization again confirms the accuracy of the voltage sensor without dependence on the light source. With these results, the justification of the measurement platform is additionally confirmed.

The summarization of the results for all four light sources is given in Table 4. The table shows the measured values in volts between the sensor and the digital multimeter. The difference is presented with the Root Mean Square Error (RMSE).

**Table 4.** The comparison of measurement results with solar light and with incandescent, CFL, and LED bulbs.

| Light Source | Sensor Avg. [V] | DMM Avg. [V] | RMSE |
|:---:|:---:|:---:|:---:|
| Solar radiation | 5.260759494 | 5.408278481 | 0.173889 |
| Incandescent bulb | 5.290892857 | 5.423053571 | 0.145102242 |
| CFL bulb | 4.311111111 | 4.431962264 | 0.113867351 |
| LED bulb | 4.401126761 | 4.510450704 | 0.115953123 |

The test proved the accuracy of the voltage sensor, justifying its usage in the proposed architecture. The difference in sensor and DMM measured values ranges from 0.11 to 0.17 V. In addition to its accuracy, the platform shows satisfying behavior during the test period. The platform was stable and easy to set up and handle.

The measurement results and the platform performance experiment can be considered successful.

*5.2. The Relation of the Other Sensor Values to the Voltage Sensor Measurements*

The experimental platform for the second phase was configured to measure the UV and sunlight intensity values, the solar panel temperature value, and the air temperature and humidity values. The specification and description of the sensors and their $R^2$ (coefficient of determination) to output voltage are given in Table 5. The coefficient of determination

$R^2$ will be used for estimation formula accuracy. Python is used for fitting the data and calculating the $R^2$. The data in the table show that the UV sensor has the greatest influence on the solar panel output voltage. The second best $R^2$ is with the visible light sensor. The reason why the UV sensor has a greater relation compared to the visible light sensor might be that the UV sensor uses 10-bit values in the range of 0–1023 to measure UV light intensity, while the BH1750 sensor is optimized for measuring light in lux units with 16-bit values ranging from 0–65535.

**Table 5.** The configuration of the experimental platform and sensor specification.

| Parameter | Sensor | Description | $R^2$ |
|---|---|---|---|
| Visible light int. | I2C BH1750 | 16-bit ADC for measuring lux | 0.94 |
| UV intensity | Analog | 10-bit ADC value range 0–1023 | 0.97 |
| Panel temperature | Analog TMP36 | 10-bit ADC value converted in °C | 0.70 |
| Air temperature | Digital DHT-11 | Value in °C | 0.60 |
| Humidity | Digital DHT-11 | Value in humidity % | N/A |

Because the UV light and BH1750FVI sensors have the highest $R^2$ values, further fitting is made with solar panel output voltage. The fitting is made with Python using exponential fitting, logarithmic, and power functions. The fitting results of the UV sensor are shown in Tables 6 and 7 for the BH1750 sensor, respectively.

**Table 6.** The fitting results for the UV sensor.

| Function | RMSE | R | $R^2$ |
|---|---|---|---|
| Linear | 0.352247 | 0.680971 | 0.463721 |
| Exponential | 0.132521 | 0.983125 | 0.966534 |
| Logarithmic | 0.326726 | 0.892513 | 0.79658 |
| Power | 0.352247 | 0.874081 | 0.764017 |

**Table 7.** The fitting results for the BH1750 sensor.

| Function | RMSE | R | $R^2$ |
|---|---|---|---|
| Linear | 31.46735 | 0.412497 | 0.1702 |
| Exponential | 0.177258 | 0.969601 | 0.940126 |
| Logarithmic | 0.494349 | 0.730967 | 0.534312 |
| Power | 0.513439 | 0.705927 | 0.498332 |

Again, the UV sensor has better fitting results than the BH1750 sensor. For both sensors, the exponential fitting function gives the best results. Three fitting functions that can be used for estimating panel output voltage based on UV sensor measurements are calculated by Python script as follows:

Exponential function

$$V_{exp} = -199.498 \cdot e^{-15.07 \cdot UV} + 5.531 \tag{1}$$

Logarithmic function

$$V_{log} = 0.37 \cdot \log(\text{UV}) + 4.747 \tag{2}$$

Power function

$$V_{pow} = 4.776 \cdot \text{UV}^{0.067} \tag{3}$$

The estimated values of output voltage calculated on the measured values of the UV sensor are given for all three functions and compared to measured voltage results and visualized (Figure 13). As in Table 6, the fitting with the exponential function provides the

best results. Two other functions (logarithmic and power) give similar results, with much lower accuracy than the exponential function.

**UV measured and estimated voltage**

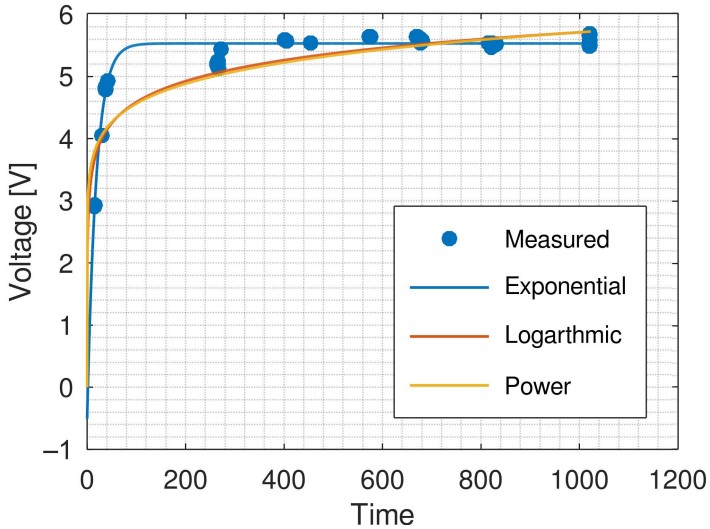

**Figure 13.** Comparison of the voltage sensor measurements and estimated values with three functions.

All three functions' fitting curves are compared to the measured output voltage values and visualized in Figure 14. Again, as shown in Table 7, the fitting with exponential functions gives the best results. Again, two other functions (logarithmic and power) provide similar results, with much lower accuracy than the exponential function. The similarity of measured values with those estimated with the exponential function is visible in Figures 13 and 14, and the values almost overlap.

**Comparison of UV fitting methods to measured voltage**

**Figure 14.** Comparison of the voltage sensor measurements and fitting curves for all three functions.

### 5.3. The Summary of the Paper's Contribution

The benefits of the proposed and existing work are multiple. This paper contributes to the distributed wireless sensor network system architecture based on open-source hardware and open-source software technologies. The proposed architecture is designed for acquiring solar radiation data and other ambient parameters. The data are collected

to define nonlinear regression estimation techniques for predicting solar panel voltage outputs and, in addition, to analyze and monitor the influence of multiple ambient data on the efficiency of solar panels. The proposed architecture is implemented at a partial scale. It is not implemented as a fully developed wireless sensor network with many sensor nodes deployed in the city area or some part of the city for long-term exploration. The proposed architecture is implemented with a prototype sensor node designed to be part of the network, as planned in this research phase. The same sensor node is evaluated for collecting solar panel voltage generation data with open-source hardware and low-cost components. The sensor node is evaluated in different scenarios with solar and artificial light conditions for the feasibility of the proposed architecture and justification of its usage. The collected data are used for creating of nonlinear regression model for voltage output estimation.

The benefits of the paper's contribution are as follows. This approach opens a way of introducing estimation techniques for sensor nodes in sensor networks. This can be useful in implementing estimation techniques in edge devices in solar data acquisition networks. Thus, we can use the sensor networks designed for usage other than solar radiation data acquisition, such as air pollution monitoring, and weather and meteorological data networks to use their sensors to predict solar radiation data. This will allow using non-specially designed wireless sensor networks for solar data estimation and solar radiation mapping, planning optimal deployment of solar-powered sensor nodes, and solar-powered sensor node efficient energy management.

Additionally, the model of estimating the solar panel performance and capacity based on the presented approach can be used to manage the energy-efficient solar-powered sensor nodes. The model for their management is an ongoing phase of the project, but it is out of the focus of this paper.

The results of this research apply to any industrial Wireless Sensor Network and the Internet of Things with solar-powered sensor nodes. This means that any outdoor deployed sensor network can use the results of this research. Those networks can be utilized in various scenarios, such as environmental monitoring, smart agriculture, the construction industry, and the smart grid [41].

Additionally, the same platform and methodology can be used in developing solar radiation estimation techniques using AI in edge computing, as seen in numerous examples [42–44]. This implementation of AI is planned as future work, and it will be based on the Python programming language and Scikit-learn [45] and Keras [46] packages.

The limitation of the study is, as mentioned before, its short evaluation period. Because of the project plan and current progress, the critical issue was to avoid long-term testing. Therefore, the prototype is tested with natural solar and artificial light in various scenarios.

## 6. Conclusions

The vital issue of designing efficient solar-powered sensor networks is addressed in this paper. A proposal for the solution to the problem is given with the architecture of the solar radiation data acquisition sensor nodes and network. The architecture of the sensor network and sensor nodes are presented and described with their main features. The proposed system collects solar radiation data and builds a model to estimate solar panel outputs accurately. The proposed platform collects data to analyze solar radiation and the influence of ambient data on solar panel performance. The collected data were used to create nonlinear regression models to estimate solar panel behavior. The created nonlinear regression models could be implemented on edge devices in complex sensor networks.

This paper proposes solar energy research and a teaching platform by combining the two approaches in solar panel research. One approach is solar radiation estimation based on various techniques using terrain, rooftop configuration, LIDAR, and other data. The different approach is the usage of Arduino-based open-source hardware components for building platforms for monitoring various solar-powered systems. The research proposes a low-cost, accurate platform for estimating solar radiation by combining these

two approaches. The key paper's contributions are a platform for data acquisition and a method for implementing estimation techniques for sensor nodes in various sensor and IoT networks, which helps achieve edge intelligence not limited to solar data acquisition networks. This system can significantly improve the process of developing systems with efficient solar powering of the sensor nodes. It can offer optimal operation settings of sensor nodes in various Industrial IoT. The small-scale prototype of the system was implemented, relying on the authors' previous experience in designing a smart factory system using open-source hardware.

The experimental results show that the system can be effectively used as a tool for data collection valid for estimating the output of sensor node-based solar panels. Thus, it can be used to determine the most suitable locations for positioning sensor nodes in various outdoor locations for Industrial IoT and other WSN, IoT, and smart city applications. Additionally, by estimating solar panel outputs, the model of the sensor node operations can be changed dynamically. Thus, during its operation, the sensor node can change its power-saving modes from full operation to modem-sleep, light-sleep, or deep-sleep modes.

The summarization of our work is as follows. This paper contributes to the proposed sensor network model for solar panel data acquisition with the models of its essential elements, such as fully-equipped and reduced-equipped sensor nodes. The model is evaluated to assess the efficiency and feasibility of using low-cost open-source hardware components to acquire solar panel data and estimate solar panel outputs. To avoid long-term tests, the experiment is performed under the simulated environment with direct solar light, reduced solar light, indoor ambient light, and different artificial light sources at different distances from the solar panel.

The evaluated sensor nodes can be used in future research for solar radiation mapping of micro-locations in urban scenarios. The solution can be used further in engineering education for building a lab for teaching IT students the development of solar-powered sensor nodes and for laboratory experiments with the solar-powered sensor node design. These features will be further explored in the future phases of this project. The other further steps in this research will be to build and validate the full-scale sensor network. The third effort in further research can be made with the collection of ambient and solar radiation data and the usage of collected data for further data analysis of ambient parameters' impact on solar panel efficiency. Together with data collection, new estimation models using different machine-learning techniques will be built. These new models will be evaluated for implementing artificial intelligence and machine learning in edge devices and, therefore, edge computing. This research direction can be extensive with the possible implementation of various AI techniques, using Python programming language and packages such as Scikit-learn and Keras and their adaptation to the Arduino platform.

**Author Contributions:** Conceptualization, D.D. and J.P.; methodology, D.D.; software, D.D.; validation, D.D., J.P., E.D. and L.R.; formal analysis, M.M. and S.M.; investigation, M.M. and S.M.; resources, D.D. and J.P.; data curation, D.D., J.P. and L.D.; writing—original draft preparation, D.D., M.M. and S.M.; writing—review and editing, E.D. and L.R.; visualization, D.D.; supervision, L.R. and I.P.; project administration, E.D.; funding acquisition, E.D. All authors have read and agreed to the published version of the manuscript.

**Funding:** The research is conducted through the project "Creating laboratory conditions for research, development, and education in the field of the use of solar resources in the Internet of Things" at the Technical Faculty "Mihajlo Pupin" Zrenjanin, financed by the Provincial Secretariat for Higher Education and Scientific Research, Republic of Serbia, Autonomous Province of Vojvodina, Project number 142-451-3118/2022-01.

**Institutional Review Board Statement:** Not applicable.

**Informed Consent Statement:** Not applicable.

**Data Availability Statement:** Not applicable.

**Acknowledgments:** This research is supported by the Provincial Secretariat for Higher Education and Scientific Research, Republic of Serbia, Autonomous Province of Vojvodina, Project number 142-451-3118/2022-01. with the project "Creating laboratory conditions for research, development, and education in the field of the use of solar resources in the Internet of Things".

**Conflicts of Interest:** The authors declare no conflict of interest.

## Abbreviations

| | |
|---|---|
| ADAS | Automatic Data Acquisition Systems |
| AI | Artificial Intelligence |
| ANN | Artificial Neural Networks |
| BP | Back Propagation |
| CFL | Compact Fluorescent Light |
| CNN | Convolutional Neural Network |
| COM | Communication port |
| CSV | Comma-Separated Values |
| DB | Database |
| DMM | Digital Multi Meter |
| GIS | Geographic Information System |
| GPU | Graphic Processing Unit |
| I2C | Inter-Integrated Circuit; pronounced as "eye-squared-C"), also I2C or IIC |
| IEEE | Institute of Electrical and Electronics Engineers |
| IIoT | Industrial Internet of Things |
| IP | Internet Protocol |
| LCD | Liquid Crystal Display |
| LED | Light-Emitting Diode |
| LIDAR | LIght Detection And Ranging |
| MCU | Micro Controller Unit |
| ML | Machine Learning |
| MQTT | MQ Telemetry Transport |
| NoSQL | non-SQL/non-relational |
| OLED | Organic Light Emitting Diode |
| OSHW | Open-Source Hardware |
| PC | Personal Computer |
| PV | Photovoltaic |
| RF | Random Forest |
| RMSE | Root-mean-square error |
| RNN | Recurrent Neural Network |
| RTC | Real-Time Clock |
| SD | Secure Digital |
| SQL | Structured Query Language |
| SVM | Support Vector Machine |
| SPI | Serial Peripheral Interface |
| TCP | Transmission Control Protocol |
| UART | Universal Asynchronous Receiver-Transmitter |
| USB | Universal Serial Bus |
| UV | Ultraviolet |
| Wi-Fi | Wireless Fidelity |
| WSN | Wireless Sensor Networks |

## Notations

| | |
|---|---|
| $V_{exp}$ | Voltage calculated with an exponential function. |
| $V_{log}$ | Voltage calculated with a logarithmic function. |
| $V_{pow}$ | Voltage estimated with a power function. |
| UV | Analog value 0–1023 of the UV sensor read |
| $R^2$ | or $r^2$ (R-square), the coefficient of determination |

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
