# Peer review of "Data Acquisition for Estimating Energy-Efficient Solar-Powered Sensor Node Performance for Usage in Industrial IoT"

_sustainability, doi:10.3390/su15097440_

Round 1

Reviewer 1 Report

  • The authors presented a distributed wireless sensor network system architecture for acquiring solar radiation data and other ambient parameters (solar panel and ambient temperature, light intensity).

  • They evaluated the proposed architecture with the prototype sensor node for collecting solar panel voltage generation data in different scenarios performed under the simulated environments.

  • According to the presented results of the experiments, the presented distributed wireless sensor network architecture can be effectively used for collecting valid data for forecasting the output of sensor node-based solar panels.

  • It is evident that the authors have previous experience in designing smart systems using an open-source-hardware.

  • The paper is well organized, and the references list is up to date. The technical correctness of the paper is good.

  • The paper is well-structured and easy to follow.

  •  

Author Response

Dear Reviewer,

Thank you very much for your time and valuable suggestions. We appreciate the time and effort you put into thoroughly analyzing our manuscript. We appreciate your commitment to the peer review process and value your opinion on the research presented in the paper.

Thank you again for your contribution to the academic community through your diligent work as a reviewer.

Kind regards,

Authors

Comments and Suggestions for Authors

The authors presented a distributed wireless sensor network system architecture for acquiring solar radiation data and other ambient parameters (solar panel and ambient temperature, light intensity).

They evaluated the proposed architecture with the prototype sensor node for collecting solar panel voltage generation data in different scenarios performed under the simulated environments.

According to the presented results of the experiments, the presented distributed wireless sensor network architecture can be effectively used for collecting valid data for forecasting the output of sensor node-based solar panels.

It is evident that the authors have previous experience in designing smart systems using an open-source-hardware.

The paper is well organized, and the references list is up to date. The technical correctness of the paper is good.

The paper is well-structured and easy to follow.

Reviewer 2 Report

 The article presents an exciting approach to address the problem of monitoring the operations of solar-powered wireless sensor nodes in outdoor Industrial IoT systems. The proposed distributed wireless sensor network system architecture based on open-source hardware and open-source software technologies appears to be a promising solution for acquiring solar radiation data and other ambient parameters crucial for achieving energy-efficient solar-powered sensor nodes' operation.

Using machine learning techniques for predicting solar panel voltage outputs is an innovative idea, and the results obtained from evaluating the prototype sensor node in different scenarios are encouraging. The article also highlights the potential of the proposed architecture for analyzing and monitoring the influence of multiple ambient data on the efficiency of solar panels.

However, some areas need to be addressed in detail.

·       The authors should provide a more detailed description of the evaluation methodology used for the prototype sensor node.

·       Additionally, more information about the scalability and adaptability of the proposed architecture for different industrial IoT environments would be helpful.

Author Response

Dear Reviewer,

 Thank you very much for your time and valuable suggestions. We appreciate the time and effort you put into thoroughly analyzing our manuscript. We appreciate your commitment to the peer review process and value your opinion on the research presented in the paper.

Thank you again for your contribution to the academic community through your diligent work as a reviewer.

We understand that clear and understandable language is necessary for effective communication. When writing the paper, I used the support of licensed software and consulted with an English translator to ensure the accuracy of the language used in the paper.

 Comments and Suggestions for Authors

 The article presents an exciting approach to address the problem of monitoring the operations of solar-powered wireless sensor nodes in outdoor Industrial IoT systems. The proposed distributed wireless sensor network system architecture based on open-source hardware and open-source software technologies appears to be a promising solution for acquiring solar radiation data and other ambient parameters crucial for achieving energy-efficient solar-powered sensor nodes' operation.

Using machine learning techniques for predicting solar panel voltage outputs is an innovative idea, and the results obtained from evaluating the prototype sensor node in different scenarios are encouraging. The article also highlights the potential of the proposed architecture for analyzing and monitoring the influence of multiple ambient data on the efficiency of solar panels.

However, some areas need to be addressed in detail.

  • The authors should provide a more detailed description of the evaluation methodology used for the prototype sensor node.

We appreciate this comment, and we provided a more detailed description of the evaluation methodology in Section 5 to make it clearer.

  • Additionally, more information about the scalability and adaptability of the proposed architecture for different industrial IoT environments would be helpful.

We appreciate this comment, and we inserted an additional explanation in the following paragraph in the newly added subsection 5.2:

“The results of this research apply to any industrial Wireless Sensor Network and the Internet of Things with solar-powered sensor nodes. This means that any outdoor deployed sensor network can use the results of this research. Those networks can be utilized in various scenarios, such as environmental monitoring, smart agriculture, the construction industry, the smart grid, etc. [41]”

Kind regards,

Authors

Reviewer 3 Report

This manuscript presents some basic information on data acquisition and assessment of solar-powered sensor node performance. It should be improved.  Following comments for authors' consideration,

(1) More details should be presented and discussed in terms of energy-efficiency estimation.

(2) Overall, the results are not analyzed in deep.

(3) Data processing is still quite entry-level.

(4) Low-cost characteristic should be clearly presented.

(5) The limitations of this study should be clearly presented.

Author Response

Dear Reviewer,

Thank you very much for your time and valuable suggestions. We appreciate the time and effort you put into thoroughly analyzing our manuscript. We appreciate your commitment to the peer review process and value your opinion on the research presented in the paper.

Thank you again for your contribution to the academic community through your diligent work as a reviewer.

Comments and Suggestions for Authors

This manuscript presents some basic information on data acquisition and assessment of solar-powered sensor node performance. It should be improved.  Following comments for authors' consideration,

3.1. More details should be presented and discussed in terms of energy-efficiency estimation.

We appreciate the comment. The explanation of why the energy-efficiency is not discussed in this paper in more detail is given in the following paragraph, inserted at the end of the newly created subsection 5.2:

“Additionally, the model of estimating the solar panel performance and capacity based on the presented approach can be used to manage the energy-efficient so-lar-powered sensor nodes. The model for their management is an ongoing phase of the project, but it is out of the focus of this paper.”

3.2. Overall, the results are not analyzed in deep.

We agree with the reviewer’s comment, and we greatly appreciate it. The deeper analyses and summarization of the results are inserted in the text in a completely new subsection 5.2 and partially in the rewritten section 5. The analyses of results are rewritten to correct the errors in the text and to make the text clearer.

The summary of the paper’s contribution and the summarized version in conclusion. 

3.3. Data processing is still quite entry-level.

We agree with the reviewer’s comment, and we greatly appreciate it.

The data processing techniques are basic and at entry-level. The paper focused not on introducing novel data processing techniques but on implementing existing techniques on limited-capacity low-cost microcontroller platforms. So, the benefit is not in the application of new but in the application of existing well-known techniques in the new platforms with limited capacity, within the new environment, and with a novel approach and purpose.

3.4. Low-cost characteristic should be clearly presented.

We agree with the reviewer’s comment, and we are thankful for helping us in improving our work. The changes with the deeper presentation of the low-cost platform are given in the added part of section 4 and the newly created Table 2.

3.5. The limitations of this study should be clearly presented.

We agree with the reviewer’s comment, and we greatly appreciate it. The limitations explanation is inserted in the text in a completely new subsection, 5.2, as the following paragraph:

“The limitation of the study is, as mentioned before, its short evaluation period. Because of the project plan and current progress, the important issue was to avoid long-term testing. Therefore, the prototype is tested with natural solar and artificial light in various scenarios.”

Kind regards,

Authors

Reviewer 4 Report

General comments

1. Define all acronyms for their first use

2. The manuscript should be revised for its grammer, typos, and flow of the content.

3. Figure 1 - quality of the figure needs improvement. it should be prepared as per the guidelines of the journal.

4. Figure 2 - Quality should be improved. Increase the Text font size. The figure is very complex in nature. It should be prepared in a simple way if possible.

5. Figure 4 - improve quality and font size

6. Figure 5 - not readable

7. line 311- check grammer. Similar changes should be made throughout the manuscript.

8. Quality of all figures should be improved.

9. A separate list of acronyms should be prepared.

10. A list of notations should be prepared.

Technical comments

1. Abstract - this paper proposes a "... network system...". Similar systems are also available in literature. How the proposed system is effective/ unique than the existing one"

2. line 30 - why the system is partially implemented and not fully. Therefore, the effectiveness of the proposed system cannot be compared with others.   

3. line 44 - what is IIOT? Define

4. Table 1 - include two more columns - 1) key outcome and 2) limitations 

5. Table 1 - The critical observations of the table should be presented (inline with the present work)

6. line 135 - the second group of "related work" should be presented in tabular form. The critical comments should also be mentioned.

7. Literature review section should be updated with more relevant papers and inline with the research gap presented.

8. The summary of the literature should be written inline with proposed work.

9. Figure 3 - rotate 90 degree (anticlockwise). Give names to significant components

10. A flowchart/ algorithm can be prepared that depcits the execution of the work.

11. Figure 8 - Figure should be modified with components names

12. Figure 10 to 12 - the critical observations and comparisons should be discussed in detail.

13. Table 3 - What are the critical learning for the parameters showing strong coorelation. The interpretations should be presented.

14. Section 5.1 - the R2 value/ RSME for linear and non-linear models should be summarized in the tabular form and then select the best-fit regression model.

15. Figure 13 and 14 - should be discussed for critical observations. 

16. Critical discussion on "the benifits of the proposed work and the existing work is missing. A separate section/ subsection should be prepared. It should clearly highlight the key contributions of the paper, key outcomes, pros and cons of the proposed work.

17. The conclusion section is weak. It should clearly presents the key outcomes of the work and outcome of comparative study. 

18. The scope of future work should be mentioned.

19. The papers cited used in this work should be relevant and from high impact factor journal. More relevant papers should be included to improve literature review and to support the problem definition. It is recommended to refer papers from all relevant/ domain specific journals.

Author Response

Dear Reviewer,

Thank you very much for your time and valuable suggestions. We appreciate the time and effort you put into thoroughly analyzing our manuscript. We appreciate your commitment to the peer review process and value your opinion on the research presented in the paper. We understand that clear and understandable language is necessary for effective communication. When writing the paper, I used the support of licensed software and consulted with an English translator to ensure the accuracy of the language used in the paper.

Thank you again for your contribution to the academic community through your diligent work as a reviewer. 

Please see the attached Response.

Round 2

Reviewer 3 Report

Thanks for the revision. Figures with high resolution should be presented.

Author Response

Dear Reviewer,

First of all, we would like to thank the  response and constructive comments and another opportunity to improve our work. We have addressed concerns raised during further editing of the manuscript and believe it has improved significantly as a result of this process. The revised manuscript is attached.

Images with a resolution of 600 dpi in Tiff format were included in the manuscript in accordance with the instructions for the authors. Image files in 600 dpi resolution, TIF format are attached.

Reviewer 4 Report

The manuscript should be updated once again based on the earlier comments

Author Response

Dear Reviewer,

First of all, we would like to thank  for response and constructive comments and another opportunity to improve our work. We have addressed concerns raised during further editing of the manuscript and believe it has improved significantly as a result of this process. The manuscript was once again revised based on earlier comments. Changes in the revised manuscript are highlighted in maroon (Round 2).

The response to the reviewer's remarks  (Round 2) can be found in the attached document.

Sincerely,

Authors. 

Round 3

Reviewer 4 Report

The manuscript addresses the comments.